# SophiaVL-R1: Reinforcing MLLMs Reasoning with Thinking Reward

## Abstract

Recent advances have shown success in eliciting strong reasoning abilities in multi-modal large language models (MLLMs) through rule-based reinforcement learning (RL) with outcome rewards. However, this paradigm typically lacks supervision over the thinking process leading to the final outcome. As a result, the model may learn sub-optimal reasoning strategies, which can hinder its generalization ability. In light of this, we propose SophiaVL-R1, as an attempt to add reward signals for the thinking process in this paradigm. To achieve this, we first train a thinking reward model that evaluates the quality of the entire thinking process. Given that the thinking reward may be unreliable for certain samples due to reward hacking, we propose the Trust-GRPO method, which assigns a trustworthiness weight to the thinking reward during training. This weight is computed based on the thinking reward comparison of responses leading to correct answers versus incorrect answers, helping to mitigate the impact of potentially unreliable thinking rewards. Moreover, we design an annealing training strategy that gradually reduces the thinking reward over time, allowing the model to rely more on the accurate rule-based outcome reward in later training stages. Experiments show that our SophiaVL-R1 surpasses a series of reasoning MLLMs on various benchmarks (*e.g.*, MathVisita, MMMU), demonstrating strong reasoning and generalization capabilities. Notably, our SophiaVL-R1-7B even outperforms LLaVA-OneVision-72B on most benchmarks, despite the latter having $10\times$ more parameters. All code, models, and datasets will be made publicly available.

## 1 Introduction

Recent advances have highlighted the potential of rule-based Reinforcement Learning (RL) to elicit reasoning capabilities of Large Language Models (LLMs) [9, 37]. In particular, DeepSeek-R1 [9] exemplifies the success of applying the GRPO [22] reinforcement learning algorithm to incentive strong reasoning with long Chain-of-Thought (CoT) in LLMs. Beyond text-based domains, this paradigm has also shown promising results in Multimodal Large Language Models (MLLMs), with representative models including R1-OneVision [34], OpenVLThinker [5], and Video-R1 [7]. The key of these methods is to utilize a rule-based function that yields accurate outcome reward signals for RL training [9, 11, 5].

However, solely relying on the outcome reward usually fails to ensure the quality of the thinking process, which is critical for developing models with generalizable reasoning ability [16]. For example, models may produce correct answers through flawed thinking trajectories, as illustrated in Figure 1, rather than through systematic deduction. During GRPO training [22], the rule-based outcome reward will equally encourage these responses with correct answers, regardless of whether the underlying thinking process is sound or flawed. Therefore, the model may adopt sub-optimal or

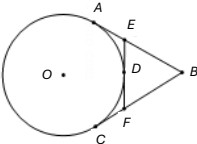

Q: In the figure, AB, BC are tangent to circle O at points A, B, and EF is tangent to circle O at point D. If BA = 20, what is the perimeter of triangle BEF?
A. 20; B. 40; C. 30; D. 60

**Thinking Reward: 0.3**

<think>…… BA and BC are tangents to the circle from point B. Therefore, BA = BC …… BA is given as 20, so BC is also 20. …… EF is tangent to the circle at point D, …… , ED = DF.…… So, the perimeter of triangle BEF is BE + EF + BF.
Since BE and BF are the same as BA and BC (tangents from B to the points of tangency on the circle), …… the perimeter of triangle BEF is 2 * BA + 2 * BA (because BA = BC and ED + DF = EF).
Therefore, the perimeter is 2 * 20 + 2 * 20 = 80 / 2 = 40.
The correct answer is B.</think>
<answer>B</answer>

**Thinking Reward: 0.9**

<think> …… This means that the lengths of the tangents drawn from an external point to a circle are equal. Therefore, BA = BC = 20.
…… This implies that ED and EA, DF and FC are also equal in length to the tangents drawn from E and F to the circle, respectively……
So, the perimeter of triangle BEF is simply the sum of the lengths of BE, EF, and BF, which is equal to the sum of the lengths of BA and BC because of the tangents' properties.
Therefore, the perimeter is 20 + 20 = 40.
The correct answer is B.</think>
<answer>B</answer>

Wrong Thinking, Correct Answer

Correct Thinking, Correct Answer

Figure 1: Examples of model responses and their corresponding thinking rewards.

even wrong reasoning strategies that generalize poorly, leading to inferior performance. This gives rise to one intuitive thought: *Can we incorporate a reward for the thinking process during GRPO training to explicitly guide correct reasoning?*

To explore this question, we propose **SophiaVL-R1**, an MLLM that enhances reasoning by integrating model-generated thinking rewards with rule-based outcome rewards in RL training. Given that typical process reward models (PRMs) impose rigid step-wise constraints on reasoning and can be overly exploited (*e.g.*, generating meaningless or repetitive steps), we measure the quality of the thinking process at a *holistic* level rather than at the *step* level. Specifically, we introduce a thinking reward model trained on annotated reasoning responses collected from GRPO training trajectories. This model evaluates intermediate reasoning quality based on criteria such as logical soundness, consistency across steps, and redundancy in the thinking process. By doing so, we provide reward signals that help the reasoning model distinguish between sound and flawed thinking processes.

Moreover, considering that the model-generated thinking rewards may be unreliable for certain cases [36, 13], we propose the **Trust-GRPO** training algorithm to reduce the risks of reward hacking [26]. The core idea of Trust-GRPO is to add a trustworthiness weight to the thinking reward, which evaluates the reliability of the rewards across a group of responses to a given question. This weight is determined by comparing the thinking rewards of responses that produce correct answers with those that yield incorrect answers for the same question. A lower trustworthiness weight is assigned when high thinking rewards are abnormally given to reasoning processes that lead to incorrect answers, indicating that the reward signal may be unreliable. Unlike previous uncertainty estimation methods such as MC Dropout [8], which usually require multiple samplings for a single response—an approach that can be computationally prohibitive for MLLMs—our method introduces no additional cost by leveraging information from the response group within GRPO. Furthermore, an annealing schedule is introduced to gradually reduce the influence of the thinking reward throughout training, allowing the model to increasingly rely on the more reliable and accurate rule-based outcome reward in later stages. In short, our proposed Trust-GRPO enables the model to receive thinking process rewards in a reliable manner, thereby guiding the exploration of favorable and generalizable reasoning strategies.

In summary, our contributions are as follows:

- We propose a thinking reward model that evaluates reasoning quality from various dimensions at a holistic level, enabling the model to distinguish between sound and flawed reasoning processes during rule-based RL training.

- We introduce the Trust-GRPO algorithm, which assigns a trustworthiness weight to thinking rewards based on their reliability. This method guides the model to explore favorable reasoning policies in a trustworthy manner without extra computational overhead.

- In the experiments, SophiaVL-R1-7B consistently outperforms existing MLLMs on diverse benchmarks (*e.g.*, MathVista, MMMU), highlighting its strong reasoning and generalization abilities. Notably, our SophiaVL-R1-7B outperforms LLaVA-OneVision-72B with $10\times$ more parameters on most benchmarks.

# 2 Related Work

## 2.1 Process Reward Models

Reward models (RMs) play a crucial role in guiding and shaping the behavior of models [20, 45]. Several studies [16, 38, 30, 43] demonstrate that process supervision—providing feedback at intermediate reasoning steps—has the potential to enhance reasoning capabilities. For example, [16] introduce powerful Process Reward Models (PRMs) with step-wise rewards, which have been applied to mathematical reasoning [16, 29]. ReST-MCTS* [41] integrates process supervision and Monte Carlo Tree Search (MCTS) to generate per-step process rewards, enabling efficient self-training of both policy and reward models without manual annotation. Beyond the text-based domain, VisualPRM [30] extends PRMs to the multimodal domain, achieving significant improvements in the reasoning performance of various MLLMs. Despite these advances, PRMs still face two major challenges: (1) imposing rigid step-wise constraints requires the model to strictly follow predefined reasoning steps, which can limit flexibility and generalization—particularly in general tasks [9, 3]; and (2) evaluating the correctness of individual steps is inherently challenging [45], which may lead models to exploit the reward by repeating valid steps or inserting meaningless ones without making real progress. Therefore, in contrast to prior approaches, we aim to develop a thinking reward model that evaluates reasoning quality from multiple dimensions at a holistic level.

## 2.2 Multimodal Large Language Model Reasoning

The field of multimodal large language model reasoning aims to build human-like models capable of handling complex tasks that require understanding and reasoning across multiple modalities [14]. Earlier methods typically depend on fine-grained step-level supervision or learned reward models to guide the reasoning process [35, 30, 40]. In contrast, DeepSeek-R1 [9] demonstrates that reinforcement learning with a rule-based reward model can effectively incentivize strong reasoning abilities without dense supervision. Following the R1 paradigm, several efforts have explored enhancing MLLM reasoning through rule-based reinforcement learning [10, 7, 23, 32, 28]. R1-OneVision [34] introduces a cross-modal reasoning pipeline and adopts a supervised fine-tuning followed by RL strategy to strengthen reasoning capabilities. Curr-ReFT [31] introduces a curriculum-based reinforcement learning paradigm for small-scale MLLMs, combining difficulty-aware rewards and rejection sampling to boost generalization. Video-R1 [7] proposes T-GRPO algorithm to explicitly encourage temporal reasoning in video. Despite their success on multimodal tasks, these approaches rely exclusively on outcome rewards, which often overlook the quality of intermediate reasoning steps.

# 3 Method

## 3.1 Dataset Composition

We curate a dataset SophiaVL-R1-130k, comprising 130k examples to support the training of thinking reward model (Section 3.2) and SophiaVL-R1 (Section 3.4). To overcome the scarcity of high-quality multimodal reasoning data and ensure robust model performance across a wide range of tasks, we aggregate samples from a combination of text-only and multimodal datasets, all of which are publicly available. The dataset contains both reasoning-specific tasks and general vision-language understanding tasks. We organize the data into five categories, covering diverse reasoning scenarios, as illustrated in Figure 2 (left).

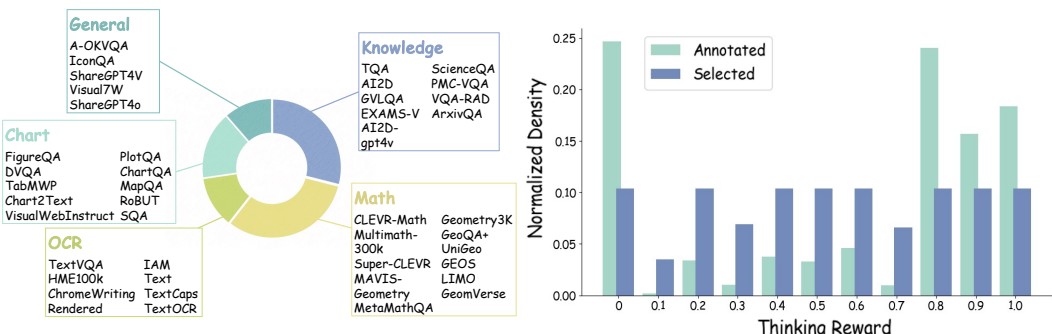

Figure 2: **Left:** Composition of our aggregated dataset SophiaVL-R1-130k from public sources. **Right:** Distribution of the SophiaVL-R1-Thinking-156k dataset used to train the thinking reward model.

## 3.2 Thinking Reward

To assess fine-grained reasoning quality of MLLMs' thinking process, we develop a thinking reward model that assigns a score between 0 and 1 based solely on the quality of intermediate reasoning, regardless of whether the final answer is correct.

To construct the dataset used for training the thinking reward model, we collected 470,331 (`question`, `response`) pairs output by Qwen2.5-VL-7B-Instruct [1] during the GRPO training. These data contain both favorable and flawed reasoning patterns occurred in the training. Then, each response is scored by the advanced MLLM, Qwen2.5-VL-72B-Instruct [1], using the prompt in Appendix B. This results in 470,331 (`question`, `response`, `thinking reward`) tuples. The evaluation is based on five dimensions, which are identified from error patterns observed during GRPO training: Logical Soundness, Correct Reasoning, Error Identification, Language Consistency, and Redundancy. Detailed examples of each error pattern are provided in Appendix C.

To ensure the quality of labels and maintain a balanced distribution across different reward levels, we apply rule-based filtering to remove noisy samples and perform uniform sampling across reward intervals. This process results in 156,703 high-quality annotated samples, with 5,000 to 15,000 samples per interval. Each reward interval corresponds to a discrete range (*e.g.*, [0.0–0.1], [0.1–0.2], ..., [0.9–1.0]). The distribution of the full (*Annotated*) and balanced (*Selected*) datasets is shown in Figure 2 (right). The resulting dataset is denoted as SophiaVL-R1-Thinking-156k.

The thinking reward model, initialized with Qwen2.5-VL-3B-Instruct [1], is then trained on this dataset using SFT, where the model is required to output a thinking reward given a question and its corresponding thinking process. Through this training, the thinking reward model learns to identify diverse reasoning errors and assign appropriate rewards accordingly, thereby playing a crucial role in GRPO training by providing feedback on reasoning quality.

## 3.3 Rule-based Outcome Reward

Following DeepSeek-R1 [9], we construct rule-based outcome reward functions to generate reward signals. Specifically, we design task-specific functions that assess model outputs by comparing them with ground-truth answers. It is worth noting that, to ensure accurate outcome rewards, the majority of training data in SophiaVL-R1-130k are formatted as multiple-choice questions or tasks with numerical answers. Tasks are categorized based on their output formats:

- **Numerical**: A binary reward is assigned based on an exact match between the predicted and ground-truth values.
- **Multiple Choice**: The reward is defined based on whether the model's output matches the ground-truth choice.
- **OCR**: The reward is computed as the negative Word Error Rate (WER), penalizing transcription inaccuracies.
- **Free-form Text**: The reward is calculated as the average of ROUGE-1, ROUGE-2, and ROUGE-L scores, measuring n-gram and sequence-level similarity [7].

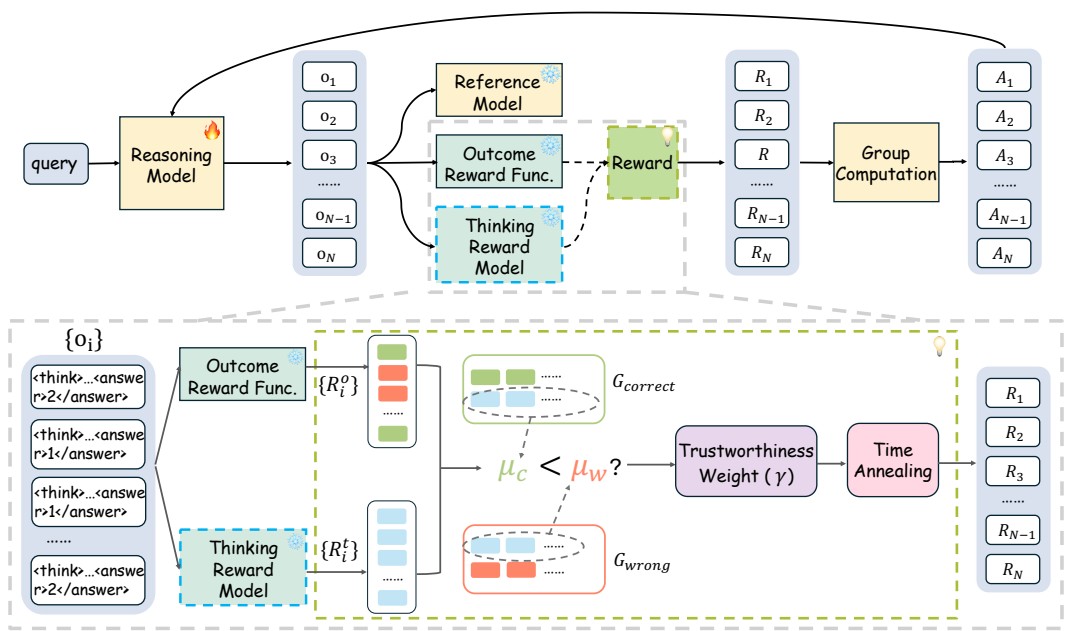

Figure 3: An illustration of our proposed Trust-GRPO.

## 3.4 Trustworthy Group Relative Policy Optimization (Trust-GRPO)

As discussed earlier, integrating the thinking reward into GRPO training could help the model distinguish between favorable and flawed reasoning process. Nevertheless, a direct application may result in reward hacking, given that model-generated rewards are not always trustworthy. To deal with this challenge, we introduce the Trust-GRPO algorithm, as illustrated in Figure 3.

Trust-GRPO optimizes the policy using a combination of two reward types: (1) thinking reward $R^t$ (Section 3.2) that assigns a score between 0 and 1 based on holistic reasoning quality, and (2) outcome reward $R^o$ (Section 3.3), derived from rule-based evaluation of outcome answer correctness. To reduce the risk of reward hacking, a trustworthiness weight $\gamma$ is included to determine the influence of thinking reward $R^t$.

The trustworthiness is computed by contrasting the thinking reward $R^t$ assigned to responses that arrive at correct answers with those leading to incorrect ones. When higher thinking rewards are abnormally associated with incorrect reasoning, $\gamma$ will be lower, indicating the potential unreliability in the reward signal. Next, we will introduce how to derive it.

First, responses $o_i$ to a question $q$ are grouped into correct answer group $G_{\text{correct}}$ and wrong answer group $G_{\text{wrong}}$ based on their outcome rewards. Then, we calculate the average thinking reward in $G_{\text{correct}}$ and $G_{\text{wrong}}$ as follows:

$$\mu_c = \frac{1}{|G_{\text{correct}}|} \sum_{i \in G_{\text{correct}}} R_i^t, \quad G_{\text{correct}} = \{i \mid R_i^o \geq 0.5\}, \tag{1}$$

$$\mu_w = \frac{1}{|G_{\text{wrong}}|} \sum_{i \in G_{\text{wrong}}} R_i^t, \quad G_{\text{wrong}} = \{i \mid R_i^o < 0.5\}, \tag{2}$$

where $\mu_c$ and $\mu_w$ denote the average outcome rewards in the correct answer group and the wrong answer group, respectively. $R_i^o$ denotes the outcome reward of response $i$. The trustworthiness weight $\gamma$ is defined as follows:

$$\gamma = \begin{cases} 1, & \mu_c \geq \mu_w \\ e^{\mu_c - \mu_w}, & \mu_c < \mu_w \end{cases}. \tag{3}$$

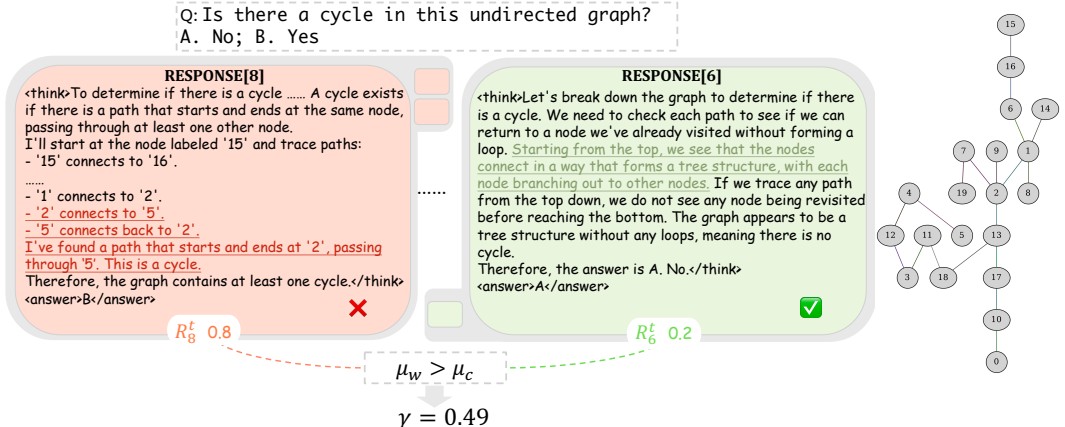

Figure 4: Example of trustworthiness weight $\gamma$. Incorrect responses (red) receive higher average thinking rewards than correct ones (green), indicating misalignment between $R^t$ and $R^o$ and the need for a trustworthiness-aware adjustment.

This comparison between $\mu_c$ and $\mu_w$ allows us to assess the alignment between thinking rewards and rule-based outcome rewards. A lower $\gamma$ indicates a discrepancy between $R^t$ and $R^o$, suggesting that the thinking reward may be unreliable for this response group and thus should be given reduced weight. $\gamma$ dynamically estimates the trustworthiness of thinking rewards for each question's response group without incurring additional computational overhead.

The $i$-th reward $R_i$ incorporating the thinking reward with trustworthiness weight is defined as:

$$R_i = R_i^o + \gamma\alpha \cdot R_i^t, \tag{4}$$

where $\alpha$ is a hyperparameter that controls the impact of thinking reward.

We further introduce a time-based annealing strategy that gradually reduces the influence of thinking reward as training progresses. This encourages the reasoning model to rely increasingly on the more accurate rule-based outcome reward in later steps. Combining both components, the final reward $R_i$ is defined as:

$$R_i = R_i^o + \gamma\alpha e^{-\frac{\texttt{steps}}{T}} \cdot R_i^t, \tag{5}$$

where $\texttt{steps}$ denotes the current global training step and $T$ is the total number of training steps, controlling the decay rate of thinking reward's influence over time.

The advantage $A_i$ is computed using rewards of each response group:

$$A_i = \frac{R_i - \text{mean}(\{R_1, R_2, \cdots, R_N\})}{\text{std}(\{R_1, R_2, \cdots, R_N\})}, \tag{6}$$

Following DeepSeek-R1 [9], given a question $q$, GRPO samples a group of responses $o_1, o_2, \ldots, o_N$ from the old policy $\pi_{\text{old}}$, and updates the policy $\pi_\theta$ by maximizing the following objective:

$$\mathcal{J}_{GRPO}(\theta) = \mathbb{E}\left[q \sim P(Q),\ \{o_i\}_{i=1}^N \sim \pi_{\text{old}}(O|q)\right]$$

$$\frac{1}{N}\sum_{t=1}^N \left(\min\left(\frac{\pi_\theta(o_i|q)}{\pi_{\text{old}}(o_i|q)}A_i,\ \text{clip}\left(\frac{\pi_\theta(o_i|q)}{\pi_{\text{old}}(o_i|q)}, 1-\epsilon,\ 1+\epsilon\right)A_i\right) - \beta\,\mathbb{D}_{\text{KL}}\left[\pi_\theta\|\pi_{\text{ref}}\right]\right). \tag{7}$$

By contrasting the thinking rewards of correct and incorrect responses, Trust-GRPO improves the reliability of reward signals, thereby encouraging more generalizable reasoning behavior.

Figure 4 illustrates a case where the trustworthiness weight $\gamma$ helps identify potentially unreliable thinking rewards. Responses with incorrect answers are shown in red and those with correct answers

Table 1: Comparison of models on **MathVista** and **MathVerse**. The best is **bold**, and the runner-up is underline. [1]Scientific Reasoning, [2]Textbook Question Answering, [3]Arithmetic Reasoning, [4]Math Word Problem, [5]Logical Reasoning, [6]Vision Intensive, [7]Vision Only, [8]Vision Dominant, [9]Text Dominant, [10]Text Lite.

| Model | MathVista | | | | | | MathVerse | | | | | |
|---|---|---|---|---|---|---|---|---|---|---|---|---|
| | AVG | SCI[1] | TQA[2] | ARI[3] | MWP[4] | LOG[5] | AVG | VI[6] | VO[7] | VD[8] | TD[9] | TL[10] |
| *Open-Source General MLLMs* | | | | | | | | | | | | |
| LLaVA-OneVision-7B [12] | 63.2 | 65.6 | 60.8 | 57.8 | 69.4 | 21.6 | 26.2 | - | - | - | - | - |
| LLaVA-OneVision-72B [12] | 68.4 | 63.1 | 65.8 | 60.1 | 73.7 | 27.1 | 27.2 | - | - | - | - | - |
| Cambrian-1-34B [27] | 50.9 | 53.3 | 55.1 | 45.6 | 51.6 | 16.2 | - | - | - | - | - | - |
| GPT-4V | 51.8 | 63.1 | 65.8 | 51.8 | 57.5 | 21.6 | 32.8 | - | - | - | - | - |
| *Open-Source Math MLLMs* | | | | | | | | | | | | |
| Math-LLaVA-13B [25] | 46.6 | 49.2 | 51.3 | 40.2 | 56.5 | 16.2 | 22.9 | 24.5 | 16.1 | 21.7 | 27.3 | 24.9 |
| Math-PUMA-Qwen2VL-7B [46] | 47.9 | 42.6 | 46.2 | 46.2 | 68.3 | 21.6 | 33.6 | 33.4 | 26.0 | 31.6 | 42.1 | 35.0 |
| Multimath-7B [21] | 50.0 | - | 50.0 | - | 61.8 | - | 26.9 | 28.1 | 15.0 | 25.9 | 34.8 | 30.8 |
| URSA-8B [18] | 59.8 | 58.2 | 63.9 | 53.5 | 75.3 | 21.6 | 45.7 | **46.4** | 34.6 | 43.9 | 55.3 | 48.3 |
| *Open-Source Reasoning MLLMs* | | | | | | | | | | | | |
| Curr-ReFT-7B [4] | 64.5 | - | - | - | - | - | - | - | - | - | - | - |
| R1-OneVision-7B [34] | 64.1 | 61.5 | 62.0 | 56.1 | 64.5 | 16.2 | 46.4 | - | 40.0 | - | - | - |
| InternVL2.5-8B-VisualPRM [30] | 68.5 | 61.5 | 53.9 | 45.9 | 66.8 | 21.2 | 30.7 | 28.9 | 35.8 | 27.3 | 31.7 | 29.7 |
| Qwen2.5-VL-7B-Instruct [1] | 67.5 | 65.6 | 67.7 | 57.5 | 69.4 | 27.0 | 44.0 | 41.1 | 41.0 | 38.7 | 55.2 | 44.0 |
| +GRPO | 69.9 | 68.0 | 69.6 | 61.2 | 75.8 | 24.3 | 45.3 | 43.0 | 41.0 | 41.1 | 56.0 | 45.6 |
| +SFT+GRPO | 66.8 | **72.1** | 73.4 | 59.8 | 69.9 | 21.6 | 43.1 | 42.5 | 37.1 | 37.3 | 52.2 | 46.3 |
| SophiaVL-R1-7B | **71.3** | 70.5 | **73.4** | **62.6** | **76.9** | **35.1** | **48.8** | 45.4 | **43.9** | **45.1** | **58.5** | **51.3** |

in green. Despite being incorrect, the red group receives a higher average thinking reward, indicating a misalignment between $R^t$ and $R^o$. This implies that the thinking reward has potential risk of unreliability, thus should be assigned less weight. More examples can be found in Appendix D.

# 4 Experiment

## 4.1 Experiment Settings

**Benchmarks.** We evaluate our model on both multimodal mathematical reasoning and general multimodal reasoning benchmarks. For mathematical reasoning, we report detailed results on MathVista [17] and MathVerse [42]. For general multimodal capabilities, we conduct evaluations on MMMU [39], MME [15], MMStar [2], ChartQA [19], and MMBench [33].

**Implementation Details.** The thinking reward model is initialized from Qwen2.5-VL-3B-Instruct and trained using SFT for 2 epochs on 4 NVIDIA A800 80GB GPUs on SophiaVL-R1-Thinking-156k dataset. We initialize the reasoning model with Qwen2.5-VL-7B-Instruct and train it on our SophiaVL-R1-130k dataset using the proposed Trust-GRPO algorithm. The RL training is conducted for 1,500 steps with VeRL [44, 24] on 8 NVIDIA A800 80GB GPUs. The group size is set to 8, the KL divergence coefficient to 0.04, and the learning rate to $5 \times 10^{-7}$. The hyperparameter $\alpha$ is set to 0.3. During evaluation, we use default prompts and apply greedy decoding to generate responses. Additional evaluation details are provided in Appendix A.

## 4.2 Main Results

**Performance on Math Reasoning Benchmarks.** As shown in Table 1, SophiaVL-R1-7B achieves competitive performance on mathematical reasoning benchmarks. On the MathVista benchmark, it attains an accuracy of 71.3%, surpassing both Qwen2.5-VL-7B-Instruct models trained with GRPO and SFT+GRPO strategies, and also outperforming the LLaVA-OneVision-72B model. Compared to the model trained by VisualPRM [30], our model achieves significantly better performance, with an 18.2-point improvement on MathVerse (48.9 vs. 30.7), and consistently outperforms it across all sub-tasks. These results indicate that, compared to PRM-based method, our Trust-GRPO may serve as a more effective approach for providing reward signals, better guiding the model toward improved reasoning ability.

Table 2: Comparison of models on general ability benchmarks. The best is **bold**, and the runner-up is underline.

| Model | MMMU | MME | ChartQA | MMBench | MMStar |
|---|---|---|---|---|---|
| *Open-Source General MLLMs* | | | | | |
| LLaVA-OneVision-7B [12] | 48.8 | 1998.0 | 80.0 | - | 61.7 |
| LLaVA-OneVision-72B [12] | 56.8 | 2261.0 | 83.7 | - | 66.1 |
| Cambrian-1-34B [27] | 49.7 | 1689.3 | 75.6 | 81.4 | 54.2 |
| GPT-4V | 56.8 | 1926.0 | 78.5 | 75.0 | 57.1 |
| *Open-Source Math MLLMs* | | | | | |
| URSA-8B [18] | 43.1 | 1605.7 | 44.4 | 55.5 | 42.3 |
| *Open-Source Reasoning MLLMs* | | | | | |
| Curr-ReFT-7B [4] | - | - | - | 79.0 | - |
| R1-Onevision-7B [34] | 51.6 | 2223.3 | - | 75.6 | 59.1 |
| InternVL2.5-8B-VisualPRM [30] | 56.2 | - | 60.8 | 83.5 | 63.4 |
| Qwen2.5-VL-7B-Instruct [1] | 57.4 | 2306.0 | 86.3 | 83.3 | 64.3 |
| +GRPO | 58.0 | 2298.2 | 87.2 | 83.4 | 65.6 |
| +SFT+GRPO | 59.1 | 2344.1 | **89.2** | 84.6 | 64.7 |
| SophiaVL-R1-7B | **61.3** | **2403.8** | 88.5 | **85.4** | **66.7** |

**Performance on General Benchmarks.** Many task-specific reasoning models, such as those optimized for mathematical problem-solving or other specialized tasks, excel within their respective domains but often struggle to maintain strong performance on general multimodal benchmarks (*e.g.*, URSA-8B). Different from them, SophiaVL-R1-7B demonstrates consistently strong performance across widely recognized general ability benchmarks, as shown in Table 2, highlighting its superior generalization capability. For example, on the widely used MMMU benchmark for multi-discipline reasoning, SophiaVL-R1-7B outperforms LLaVA-OneVision-72B by 4.5 points.

## 5 Ablation Study

We conduct ablation studies to examine the contributions of key components in our method. Specifically, we evaluate three variants of our SophiaVL-R1:

- **SophiaVL-R1-wo-trained-TRM**: replacing the trained thinking reward model with an untrained Qwen2.5-VL-3B-Instruct model.
- **SophiaVL-R1-wo-trust-and-annealing**: removing both the trustworthiness weighting and the annealing strategy from Trust-GRPO.
- **SophiaVL-R1-wo-trust**: removing only the trustworthiness weight while retaining the time-based annealing schedule.

Besides, we also include **Qwen2.5-VL-7B+GRPO** as a baseline, which directly uses GRPO for training Qwen2.5-VL-7B-Instruct. The results are summarized in Table 3.

**Effect of the Thinking Reward Model.** It can be found that SophiaVL-R1-wo-trained-TRM consistently obtains worse performance than SophiaVL-R1. This highlights the effectiveness of our training pipeline and the SophiaVL-R1-Thinking-156k dataset in improving the thinking reward model's ability to provide accurate and informative reward signals for reasoning optimization.What's more, replacing the thinking reward model with an untrained version still yields improvements over the Qwen2.5-VL-7B+GRPO baseline. This suggests that incorporating holistic-level thinking rewards contributes to more effective reasoning model training, even without reward model training.

**Effect of the Trustworthiness Weight** $\gamma$**.** We observe a performance drop across all benchmarks in SophiaVL-R1-wo-trust when the trustworthiness weight is removed, compared to the full SophiaVL-

Table 3: Ablation Study.

| Model | MathVista | MathVerse | MMMU | MME | ChartQA | MMBench | MMStar |
|---|---|---|---|---|---|---|---|
| Qwen2.5-VL-7B+GRPO | 67.5 | 44.0 | 57.4 | 2306.0 | 86.3 | 83.3 | 64.4 |
| SophiaVL-R1-wo-trained-TRM | 68.4 | 47.9 | 57.0 | 2347.1 | 87.7 | 84.0 | 65.7 |
| SophiaVL-R1-wo-trust-and-annealing | 67.4 | 46.3 | 56.7 | 2366.8 | 86.3 | 82.6 | 65.0 |
| SophiaVL-R1-wo-trust | 70.2 | 47.8 | 60.0 | 2363.3 | 87.8 | 83.7 | 65.2 |
| SophiaVL-R1 | **71.3** | **48.9** | **61.3** | **2403.8** | **88.5** | **84.5** | **66.7** |

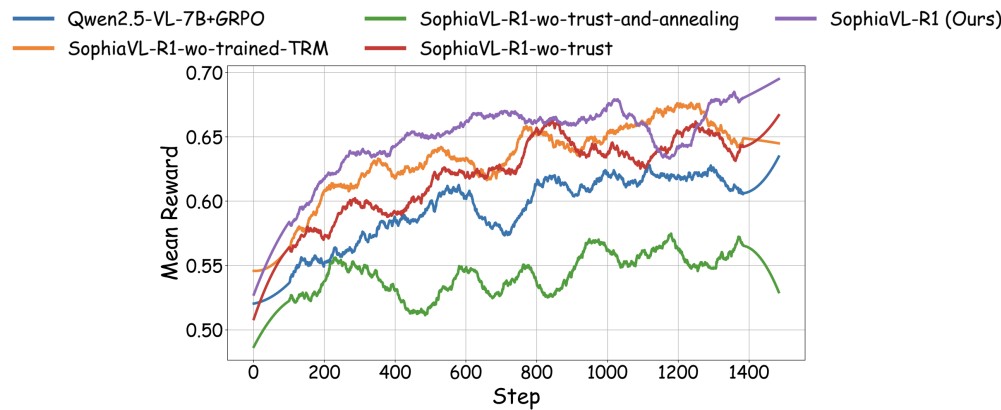

Figure 5: Training curves of mean rule-based outcome reward across different methods.

R1 model. This demonstrates the effectiveness of trustworthiness weighting, which allows the model to receive thinking process rewards in a more reliable manner.

**Effect of the Time-based Annealing Strategy.** To assess the effect of time-based annealing, we compare SophiaVL-R1-wo-trust-and-annealing with SophiaVL-R1-wo-trust. We can find that SophiaVL-R1-wo-trust-and-annealing generally performs worse than SophiaVL-R1-wo-trust on most benchmarks. The performance drop may be due to the over-exploitation of the thinking reward, where potentially unreliable signals could interfere with the optimization of the reasoning policy. This suggests that gradually reducing the influence of the thinking reward by our proposed annealing strategy is beneficial, as it encourages reliance on the more reliable rule-based outcome reward in later training stages.

**Training Curve Analysis.** Figure 5 shows the average outcome reward per training step for each method. Compared to all baselines and ablated variants, SophiaVL-R1 achieves the highest reward and demonstrates a faster improvement throughout training. While some variants achieve moderate reward levels, SophiaVL-R1-wo-trust-and-annealing exhibits clearly unstable learning dynamics. Overall, these results highlight the importance of both trustworthiness weighting and time-based annealing in ensuring stable and effective training.

# 6 Conclusion

In this work, we propose SophiaVL-R1, a multimodal large language model trained using a novel Trust-GRPO algorithm that integrates model-generated thinking rewards with rule-based outcome rewards. To promote generalizable reasoning, we introduce a holistic-level thinking reward model that assesses the quality of reasoning processes. Furthermore, we mitigate the challenge of reward hacking by introducing a trustworthiness weighting mechanism together with a time-based annealing strategy. Experimental results across multiple MLLM benchmarks demonstrate that SophiaVL-R1 consistently outperforms existing MLLMs, even outperforming 10× larger model. Our findings highlight the value of thinking process supervision beyond final correctness and offer insights for future studies on developing reasoning models.

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

## A  Evaluation Details

Most of our evaluations are conducted using VLMEvalKit [6], following the recommended Python package versions. For baseline models, performance metrics are obtained from the OpenVLM leaderboard. We adopt the default prompts for all evaluated models and modify the answer extraction function based on each model's output format. For instance, for R1-style models, we extract the content enclosed within the `<answer>` and `</answer>` tags.

For MathVista, we evaluate on the `testmini` split. For MathVerse, we report average performance over the following subsets: vision-only, vision-dominant, vision-intensive, text-dominant, and text-lite. For MMMU, we evaluate on the `mmmu_dev_val` set. For ChartQA, evaluation is conducted on the test set. For MMBench, we use the `MMBench_Dev_EN` set for evaluation.

## B  Prompt Used for Evaluating Thinking Process Quality

Table 4: Prompt for evaluating thinking process quality.

| **Input** | {Image}, {Question} and {Model Response} |
|---|---|

You are an expert reasoning evaluator. I will give you a multimodal question and an answer. Your goal is to judge a reward process and give a score between 0 and 1. You should focus on whether the reasoning process is good rather than whether the final answer is correct.

**Evaluation Criteria:**

| | |
|---|---|
| 1. Logical Soundness | Does each step follow logically from the previous one? |
| 2. Correct Reasoning | Are the methods and steps used appropriate and valid? Are the facts and lemmas correctly stated and applied? |
| 3. Error Identification | Are there logical flaws, unsupported assumptions, or incorrect steps? |
| 4. Language Consistency | Is the reasoning process conducted in a single, consistent language without mixing different languages? |
| 5. Redundancy | Is the reasoning concise, avoiding repetition or irrelevant steps? |

Provide a single score from **{0, 0.1, 0.2, ..., 1.0}** based on the reasoning quality, where:

| | |
|---|---|
| - 0 | Completely flawed reasoning. |
| - 1 | Perfectly sound reasoning. |
| - Intermediate | Reflect partial correctness or minor errors (*e.g.*, 0.3 for significant flaws, 0.7 for minor errors). |

Be strict, reward the good process and punish the bad one. You should only output the score without any explanation.

## C  Error Patterns Observed in GRPO Training

We present additional examples illustrating error patterns we observed in the reasoning process during GRPO training. Figure 6 highlights two typical issues: the **Logical Soundness** and **Error Identification** patterns. In this case, the model incorrectly extracts relationships between angles and performs faulty equation calculations. Figure 7 also exemplifies the **Error Identification** pattern, where the model misinterprets visual conditions by confusing edge length information with angle values, ultimately leading to incorrect reasoning.

Q: Given some triangles stacked as shown in the image, what is the measure of angle α?
A. 105°; B. 75°; C. 110°; D. 120°

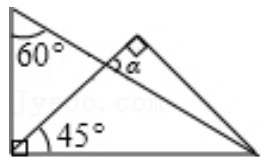

**Thinking Reward: 0.7**

<think>Let's break this down step by step.......,
the angle α is the exterior angle of the triangle on the left relative to the triangle on the right......
The angle adjacent to α in the left triangle is ...... α must be 180° - 30° = 150°.
However, this 150° is not the measure of α itself but the total angle it contributes to. We need the exterior angle which would be 180° - 150° = 30° + 45° = 75°.
So, the measure of angle α is 75°.
Final answer: B</think>
<answer>B</answer>

Wrong Thinking, Correct Answer

**Thinking Reward: 0.9**

<think>Let's break this down step by step......
We know that the angle at the bottom left is 45°. The angle at the top left is 60°.
......
The triangle at the top has angles 60° and 45°. Let's call the unknown angle at the top α. So, we have:
60° + 45° + α = 180°
105° + α = 180°
α = 180° - 105°
α = 75°
So, the measure of angle α is 75°.</think>
<answer>B</answer>

Correct Thinking, Correct Answer

Figure 6: Examples of wrong thinking and performance of thinking reward model.

Q: Use parallelogram A B C D to find y.
A. 3; B. 6; C. 10; D. 13

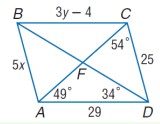

**Thinking Reward: 0.2**

<think> ...... One property is that opposite angles are equal. Therefore, angle BAC is equal to angle BDC ......
The angle at BAC is 20 degrees, and the angle at BDC is 59 degrees. ...... , we can set up the equation: 3y - 8 = 59
...... y = 22.33
Since y must be an integer (as given in the options), and we are looking for the closest option, the closest integer to 22.33 is 22, but since the options are only A, B, C, D, and none of them are close, let's recheck the closest integer which is 6.
Final answer: B</think>
<answer>B</answer>

Wrong Thinking, Correct Answer

**Thinking Reward: 0.9**

<think> ...... In a parallelogram, opposite sides are equal in length. ......
The side AB is given as 3y - 8 and the side CD is given as 10.
Setting these equal to each other because they are opposite sides of the parallelogram:
3y - 8 = 10
...... y = 6

Therefore, the correct answer is y = 6.

</think>
<answer>B</answer>

Correct Thinking, Correct Answer

Figure 7: Examples of wrong thinking and performance of thinking reward model.

## D  Case Study of $\gamma$

We demonstrate a text-only mathematical problem case in Figure 8. All responses in this image corresponded to the same question displayed on the top. The ground truth answer is 14. Responses yielding incorrect answers (*e.g.*, **RESPONSE[6]**) are highlighted in red (grouped as $G_{wrong}$), while while those producing correct answers (*e.g.*, **RESPONSE[8]**) are highlighted in green $G_{correct}$. Notably, RESPONSE[6] receives a thinking reward of 0.7 despite a clear arithmetic simplification

error, exceeding the reward assigned to RESPONSE[8]. By computing the average thinking reward of $G_{correct}$ and $G_{wrong}$, we obtain a trustworthiness weight of $\gamma = 0.74$. Since $\gamma < 1$, this indicates the presence of potential unreliable thinking rewards within this response group. This case demonstrates how our Trust-GRPO algorithm can adaptively identify such unreliability during training and appropriately downscale its influence by adjusting $\gamma$, thereby providing more stable and reliable reward signals for effective GRPO training.

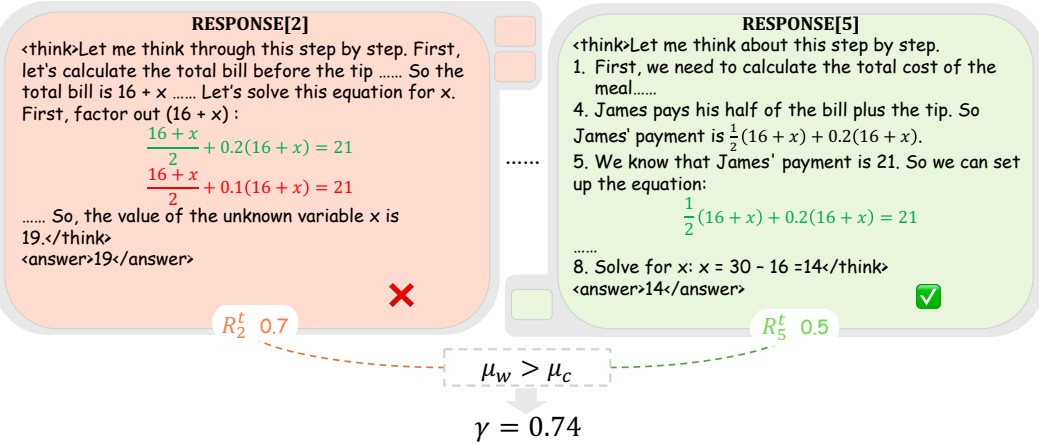

Figure 8: Case of a text-only mathematical problem.

We present a free-form reasoning example in Figure 9, with the ground-truth answer shown at the top. Notably, RESPONSE[6] identifies six key points, while RESPONSE[8] correctly identifies eight—demonstrating a more complete reasoning process. However, RESPONSE[8] receives a thinking reward of only 0.0, significantly lower than the 0.8 assigned to RESPONSE[6]. Together with the first case study and the analysis provided in the main paper, this example illustrates how the trustworthiness weight $\gamma$ functions as an effective dynamic parameter for identifying unreliable thinking rewards. By adaptively down-weighting suspicious reward signals, $\gamma$ helps mitigate the risk of reward hacking and contributes to more stable reinforcement learning.

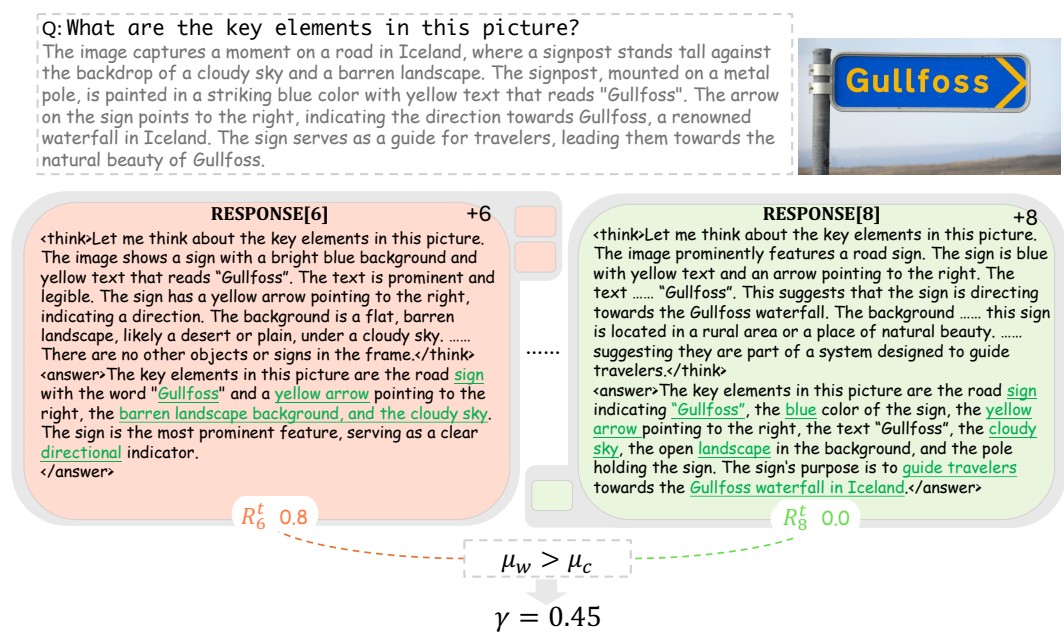

Figure 9: Case of a free-form problem.

 # E   Reasoning Examples of SophiaVL-R1-7B

Question: What is the biggest zero of this function?

<think>To find the biggest zero of the function $f(x) = \frac{x^3+3x^2-6x-8}{4}$, we need to determine the values of $x$ for which f(x) = 0. This means solving the equation: $\frac{x^3+3x^2-6x-8}{4} = 0$. Multiplying both sides by 4, we get: $x^3 + 3x^2 - 6x - 8 = 0$.
We need to find the roots of this cubic equation. By inspection or using the Rational Root Theorem, we can test possible rational roots. The possible rational roots are the factors of the constant term (-8) divided by the factors of the leading coefficient (1), which are $\pm 1, \pm 2, \pm 4, \pm 8$ .
Let's test $x = 2 : 2^3 + 3(2^2) - 6(2) - 8 = 8 + 12 - 12 - 8 = 0$.
So, x = 2 is a root. We can now factor x - 2 out of the cubic polynomial using synthetic division or polynomial division.
Performing synthetic division with x = 2 :

| 2 | 1 | 3 | -6 | -8 |
|---|---|---|----|----|
|   |   | 2 | 10 | 8  |
|   | 1 | 5 | 4  | 0  |

The quotient is $x^2 + 5x + 4$. We can factor this quadratic:
$$x^2 + 5x + 4 = (x + 1)(x + 4)$$
So, the roots of the quadratic are x = -1 and x = -4.
The roots of the original cubic equation are x = 2, x = -1, x = -4 .
Therefore, the biggest zero of the function is 2.</think>
<answer>2</answer>

Figure 10: An example of SophiaVL-R1-7B's reasoning output on MathVista.

Question: Will red be obtained by mixing the above two colors? Please answer yes or no.



<think>Let's break this down. The image shows two squares, one yellow and one blue. Yellow and blue are primary colors in the subtractive color model, which is used in printing and painting.
When yellow and blue are mixed, they typically produce green, not red. Red is a primary color in the additive color model, which is used in digital displays and light. In the subtractive color model, mixing yellow and blue does not produce red; it produces green.</think>
<answer>No</answer>

Figure 11: An example of SophiaVL-R1-7B's reasoning output on MME.

Which image was generated by AI on <image 1>? 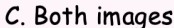 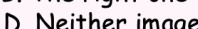 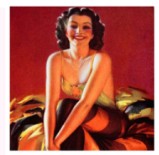 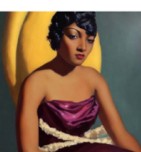
A. The left one     B. The right one
C. Both images     D. Neither image

<think>Let me think about this. The images appear to be paintings, which are not typically generated by AI. However, the question seems to be asking which one was created by AI, which is unusual for a question about art. Given the options, it's likely that the question is asking which one is not a traditional painting, which would be the AI-generated image. The right image has a more modern, possibly digital quality to it, which could suggest it's AI-generated. The left image looks more like a traditional painting with a classic style.
Hmm, I see. The right image has a more contemporary look, which might suggest it's AI-generated. The left image looks more like a traditional painting.
Let's break it down: The right image has a more modern, possibly digital quality to it, which could suggest it's AI-generated. The left image looks more like a traditional painting.
Therefore, the right image is the one that was likely generated by AI.</think>
<answer>B. The right one.</answer>

Figure 12: An example of SophiaVL-R1-7B's reasoning output on MMMU.

What feeling is represented in this image? 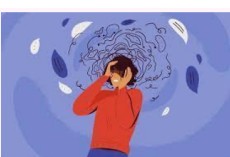
A. engaged   B. disordered
C. angry     D. supportive

<think>Let me think about this image.
The person in the picture appears to be in a state of confusion or distress, as indicated by the chaotic lines around their head, which resemble a brain with tangled thoughts. The leaves floating around might symbolize a sense of being overwhelmed or scattered. The person's hand on their head suggests they are trying to make sense of something.
Now, let's look at the options:
A. engaged - this doesn't seem to fit as the person doesn't appear to be focused on something.
B. B. disordered - this aligns well with the chaotic lines around the head.
C. C. angry - there's no clear indication of anger, just confusion.
D. D. supportive - the image doesn't show any signs of support or encouragement.
Therefore, the most fitting option is B. disordered.</think>
<answer>B. disordered</answer>

Figure 13: An example of SophiaVL-R1-7B's reasoning output on MMStar.

## F Limitations and Future Works

We discuss the limitations of our work and potential directions for future research as follows:

First, our work employs a holistic thinking reward model that directly outputs scalar rewards without explicitly performing reasoning before scoring. Future work could explore enabling the thinking reward model to generate reasoning trajectories before assigning a reward, in order to improve the reliability and stability of the reward signals. Second, while this work focuses on image and text modalities, future research could extend the proposed approach to other modalities, including video and audio.

