# OpenReview forum: "SophiaVL-R1: Reinforcing MLLMs Reasoning with Thinking Reward"
_NeurIPS.cc/2025/Conference — Submitted to NeurIPS 2025_

### Official Review · Reviewer_Qfk6 · 2025-06-19

**Clarity:** 3
**Significance:** 2
**Originality:** 3
**Rating:** 4
**Confidence:** 4

**Summary:**

This paper introduces an improved Trust-GRPO method to enhance the reasoning ability of MLLMs. The authors argue that the rule-based RL with outcome rewards focuses solely on the final answer, neglecting the quality of the intermediate reasoning trajectory. To address the limitation, they propose incorporating a thinking reward model in addition to the conventional GRPO. Experiments on VL reasoning and generalization benchmarks indicate that the proposed method achieves better performance.

**Questions:**

1. In Table 1 and Table 2, the GPT-4V is categorized under Open-Source General MLLMs, and I do not think it is proper.

2. The construction of SophiaVL-R1-Thinking-156k includes Gereral and Math data, which appear to overlap in domain with the evaluation benchmarks. Have the authors taken measures to ensure that there is no data leakage?

3. Does the SFT thinking reward model assign a high score to responses that including the correct final answer but incorrect reasoning process?

**Ethical Concerns:**

["NO or VERY MINOR ethics concerns only"]

**Limitations:**

Yes

**Quality:**

3

**Strengths And Weaknesses:**

## Strengths:

1. The incorporation of a thinking reward model alongside traditional GRPO represents a novel approach to reinforcing intermediate reasoning quality, rather than just outcomes.
2. The proposed method demonstrates improved results on vision-language reasoning and generalization benchmarks, indicating its practical effectiveness.

## Weaknesses:

1. The created dataset, SophiaVL-R1-Thinking-156k, is annotated using a larger VL model; however, there is no detailed explanation of the quality assurance process or validation methodology for the annotations.

2. The thinking reward model (Qwen2.5-VL-3B-Instruct) is trained via SFT on the created dataset. However, there is no experiment to evaluate the accuracy or reliability of the generated reward scores.

3. Since the thinking reward model is trained on SFT data annotated by a larger VL model (Qwen2.5-VL-72B-Instruct), it is unclear why not choose to directly use the larger VL model as the reward model. The paper does not provide a rationale for this design.

4. The method introduces several hyperparameters (e.g., the value 0.5 in Equations 1 and 2, the exponential function in Equation 3, and the alpha in Equation 4), which seem arbitrary and are not well justified in the paper.

---

> ### Author Rebuttal · Authors · 2025-07-30
>
> > Q1: The created dataset, SophiaVL-R1-Thinking-156k, is annotated using a larger VL model; however, there is no detailed explanation of the quality assurance process or validation methodology for the annotations.
>
> A1: Thanks for your comments. To improve annotation quality in the SophiaVL-R1-Thinking-156k dataset, we adopt a structured scoring protocol based on five criteria (Line 121–128 and Appendix B), which reflect common error types we observed during GRPO training. This design guides the reward model to assess COT quality, rather than relying solely on the correctness of the final answer.
>
> To evaluate the quality of our annotation,  we randomly sample 3000 examples from the SophiaVL-R1-Thinking-156k dataset and rescore them using GPT-4o under the same prompt settings. **The** **Spearman** **correlation between our annotated reward scores and those from GPT-4o is 0.73**. Since the Spearman coefficient ranges from –1 to 1, this result indicates **strong agreement in** **annotation** **behavior**.
>
> |                                                              | Spearman correlation |
> | ------------------------------------------------------------ | -------------------- |
> | Our annotation by Qwen2.5-VL-72B-Instruct vs. GPT-4o's annotation | 0.73                 |
>
> This level of alignment provides an external reference supporting the reliability of our annotations. We will explore more effective methods to improve and ensure annotation quality in future work.
>
> > Q2: The thinking reward model (Qwen2.5-VL-3B-Instruct) is trained via SFT on the created dataset. However, there is no experiment to evaluate the accuracy or reliability of the generated reward scores.
>
> A2: Thanks for your comments. To evaluate the reliability of our SFT-trained thinking reward model, we conducted a correlation analysis on 3000 randomly sampled examples, similar to previous anwer. We compared the reward scores from the thinking reward model with annotations from GPT-4o.
>
> |                                                              | Spearman correlation |
> | ------------------------------------------------------------ | -------------------- |
> | Qwen2.5-VL-3B-Instruct's annotation vs. GPT-4o's annotation  | 0.39                 |
> | Our trained thinking reward model's annotation vs. GPT-4o's annotation | 0.55                 |
>
> The Spearman correlation coefficient between the thinking reward model and GPT-4o is 0.55, which improves upon the 0.39 correlation of the original Qwen2.5-VL-3B-Instruct model. This shows that SFT enhances the alignment of the reward model with higher-quality annotations, supporting its reliability.
>
> > Q3: Since the thinking reward model is trained on SFT data annotated by a larger VL model (Qwen2.5-VL-72B-Instruct), it is unclear why not choose to directly use the larger VL model as the reward model. The paper does not provide a rationale for this design.
>
> A3: Thanks for your comments. The main reason we use a smaller reward model is to **reduce computational overhead**. Unlike prior methods such as DPO, GRPO generates multiple responses per question (typically more than 8). Each response requires a separate forward pass through the reward model. Using a large reward model like Qwen2.5-VL-72B-Instruct in GRPO training would significantly increase training costs.
>
> Moreover, our results (Table 1 and Table 2) show that our 3B reward model leads to clear improvements on reasoning benchmarks, suggesting that it is both efficient and effective. We will explore larger reward models in the future.
>
> > Q4: The method introduces several hyperparameters (e.g., the value 0.5 in Equations 1 and 2, the exponential function in Equation 3, and the alpha in Equation 4), which seem arbitrary and are not well justified in the paper.
>
> A4: Thanks for your comments. The hyperparameters in our method are selected empirically. Due to the high computational cost of training MLLMs, it is not feasible for us to conduct extensive sensitivity analysis on each hyperparameter. We plan to explore more systematic hyperparameter tuning in future work when resources permit.
>
> > Q5: In Table 1 and Table 2, the GPT-4V is categorized under Open-Source General MLLMs, and I do not think it is proper.
>
> A5: Thanks for pointing this out. We will correct these issues in the final version.
>
> > Q6: The construction of SophiaVL-R1-Thinking-156k includes Gereral and Math data, which appear to overlap in domain with the evaluation benchmarks. Have the authors taken measures to ensure that there is no data leakage?
>
> A6: Thanks for your comments. To prevent data leakage, we carefully use only the training splits of public datasets when constructing SophiaVL-R1-Thinking-156k. Since evaluation benchmarks typically rely on validation or test splits, or on newly annotated samples outside public training data, this ensures there is no overlap between our training data and evaluation sets.
>
> > Q7: Does the SFT thinking reward model assign a high score to responses that including the correct final answer but incorrect reasoning process?
>
> A7: Thanks for your comments. The thinking reward model generally assigns lower scores to responses that have correct final answers but incorrect reasoning. We manually evaluate 60 responses, with 30 examples each of (1) correct final answers but incorrect reasoning, and (2) both correct final answers and correct reasoning.
>
> The average thinking reward was 0.34 for the first group and 0.78 for the second, showing that the model significantly penalizes incorrect reasoning even though the final answer is correct.
>
> |                                               | Thinking Reward |
> | --------------------------------------------- | --------------- |
> | correct final answers but incorrect reasoning | 0.34            |
> | correct final answers and correct reasoning   | 0.78            |

---

### Official Review · Reviewer_XUT8 · 2025-06-23

**Clarity:** 3
**Significance:** 3
**Originality:** 3
**Rating:** 5
**Confidence:** 5

**Summary:**

This paper proposes SophiaVL-R1, a MLLM that enhances reasoning ability by combining GRPO with a thinking reward evaluating the quality of reasoning steps. To address possible unreliability in thinking rewards, the authors introduce a Trust-GRPO algorithm that dynamically adjusts the influence of the thinking reward during training. They also construct SophiaVL-R1-130k dataset for GRPO training and SophiaVL-R1-Thinking-156k dataset for reward model training. The model is tested on several benchmarks and shows strong performance.

**Questions:**

In Figure 5, the SophiaVL-R1 model achieves higher rewards compared to the GRPO baseline. Is this improvement primarily due to the inclusion of the additional thinking reward?

**Ethical Concerns:**

["NO or VERY MINOR ethics concerns only"]

**Final Justification:**

After the rebuttal and discussion, the authors’ clear description of the data composition and evaluation of the intermediate reasoning process has fully addressed my concerns. Therefore, I have raised my score from 4 to 5.

**Limitations:**

Yes

**Paper Formatting Concerns:**

I don't notice any major formatting issues.

**Quality:**

3

**Strengths And Weaknesses:**

Strength
1. The authors introduce a reward model into the GRPO, which improves the model’s reasoning capabilities.
2. The paper demonstrates strong generalization ability, particularly on the MMMU benchmark.
3. The authors construct a point-wise reward dataset for reasoning evaluation, which can be valuable for further research in this area.

Weaknesses
1. The reward model used for assessing the reasoning process is likely to suffer from significant hallucination, as observed even in very large models (such as 72B or proprietary models). The paper uses a 3B reward model, but does not provide quantitative metrics (e.g., on VLRewardBench or other benchmarks) to support its reliability.  Considering that this paper adopts a point-wise reward model, the authors could directly evaluate the reward model’s ability to distinguish between “chosen” and “rejected” responses by comparing the assigned scores for each.
2. Although the paper shows strong results on some benchmarks (e.g., MMMU), the improvement on MathVista and MathVerse is relatively modest, especially given the large amount of training data used. The authors should provide ablation results with varying data scales (e.g., using 50% or 25% of the data), to better clarify its effectiveness.
3. The paper lacks detailed discussion about the sources and distribution of SophiaVL-R1-Thinking-156k dataset.
4. While the paper uses a reward model to evaluate the thinking process, there is a lack of direct assessment of the intermediate reasoning steps, such as evaluation on benchmarks like MME-CoT.

---

> ### Author Rebuttal · Authors · 2025-07-30
>
> > Q1: The reward model used for assessing the reasoning process is likely to suffer from significant hallucination, as observed even in very large models (such as 72B or proprietary models). The paper uses a 3B reward model, but does not provide quantitative metrics (e.g., on VLRewardBench or other benchmarks) to support its reliability. Considering that this paper adopts a point-wise reward model, the authors could directly evaluate the reward model’s ability to distinguish between “chosen” and “rejected” responses by comparing the assigned scores for each.
>
> A1: Thanks for your comments. Our motivation for using a compact 3B reward model lies in its training efficiency in GRPO. Unlike prior methods such as DPO, GRPO generates multiple responses per question (typically more than 8). Each response requires a separate forward pass through the reward model. Using a large reward model (e.g., 72B) in GRPO training would significantly increase training costs.
>
> To further evaluate the reliability of our reward model, we report its performance on VLRewardBench. As shown below, our 3B thinking reward model achieves performance comparable to that of GPT-4o-mini and Qwen2-VL-72B.
>
> |                                | General | Hallucination | Reasoning | Overall Accuracy | Macro Accuracy |
> | ------------------------------ | ------- | ------------- | --------- | ---------------- | -------------- |
> | GPT-4o-mini                    | 41.7    | 34.5          | 58.2      | 41.5             | 44.8           |
> | Qwen2-VL-72B                   | 38.1    | 32.8          | 58.0      | 39.5             | 43.0           |
> | Our Thinking Reward Model (3B) | 44.8    | 45.4          | 50.9      | 46.7             | 47.1           |
>
> We will add the VLRewardBench results in the final version.
>
> > Q2: Although the paper shows strong results on some benchmarks (e.g., MMMU), the improvement on MathVista and MathVerse is relatively modest, especially given the large amount of training data used. The authors should provide ablation results with varying data scales (e.g., using 50% or 25% of the data), to better clarify its effectiveness.
>
> A2: Sorry for confusing you. We would like to clarify that we **did not use the entire dataset** for training. Due to the high computational cost of GRPO—where each question typically requires more than 8 sampling—many prior works, such as UI-R1 [1], Visual-RFT [2], and Search-R1 [3], also train with a limited number of steps (often around 1000).
>
> Following this common practice, we trained for only **1500 steps** on 8 A800 GPUs, which already takes about **2 days** and uses approximately **10% of the full dataset**. We believe that scaling up training could further improve performance and leave this exploration to future work.
>
> [1] UI-R1: Enhancing Efficient Action Prediction of GUI Agents by Reinforcement Learning
>
> [2] Visual-RFT: Visual Reinforcement Fine-Tuning
>
> [3] Search-R1: Training LLMs to Reason and Leverage Search Engines with Reinforcement Learning
>
> > Q3: The paper lacks detailed discussion about the sources and distribution of SophiaVL-R1-Thinking-156k dataset.
>
> A3: Thanks for your comments. In the final version, we will include a figure that clearly illustrates the dataset composition details.
>
> > Q4: While the paper uses a reward model to evaluate the thinking process, there is a lack of direct assessment of the intermediate reasoning steps, such as evaluation on benchmarks like MME-CoT.
>
> A4: Thanks for your comments. Due to the time constraints in the rebuttal, we plan to evaluate the performance of our model on MME-CoT in the future and add the corresponding results in the final version.
>
> > Q5. In Figure 5, the SophiaVL-R1 model achieves higher rewards compared to the GRPO baseline. Is this improvement primarily due to the inclusion of the additional thinking reward?
>
> A5: Sorry for confusing you. In Figure 5, the **y-axis shows the mean rule-based outcome reward only**, **without including the thinking reward**. As illustrated in the figure, SophiaVL-R1 generally achieves both faster reward improvement and a higher final outcome reward compared to the GRPO baseline. Moreover, across all ablation variants, SophiaVL-R1 generally obtains higher outcome rewards, indicating the effectiveness of our method.

---

> > ### Comment · Reviewer_XUT8 · 2025-08-04
> >
> > Thank you for the authors’ response. However, **most of my concerns remain unaddressed.**
> >
> > Q2. As far as I know, the authors did not state in the paper that only 10% of the data was used for training, which differs from the paper. How did the authors sample this 10% of the data? Would using more data lead to improved performance?
> >
> > Q3. Are SophiaVL-R1-130k and SophiaVL-R1-Thinking-156k derived from the same source data? Could the authors provide a textual explanation or a table to clarify the data sources and composition of the SophiaVL-R1-Thinking-156k dataset?
> >
> > Q4. It is also important to provide an evaluation of the process quality, especially since the authors used a reward model to supervise the intermediate reasoning steps. Moreover, I believe that evaluating on MME-CoT would not be time-consuming.
> >
> > Q5. Since Figure 5 only includes rule-based outcome reward, why do the initial performance levels of the methods shown in the figure not start from the same baseline?

---

> ### Author Response · Authors · 2025-08-05
>
> Thank you for your feedback. We would like to address your remaining concerns point by point below.
>
> > Q2. As far as I know, the authors did not state in the paper that only 10% of the data was used for training, which differs from the paper. How did the authors sample this 10% of the data? Would using more data lead to improved performance?
>
> A2: Sorry for confusing you. As stated in **Lines 207–208** of our manuscript, we trained the model for **1,500 steps** on 8 GPUs, with each GPU processing one sample per step (we apologize for omitting the batch size). This setup results in **1,500 × 8 = 12,000** training samples in total, which corresponds to approximately **10%** of the full SophiaVL-R1-130k dataset. These data are **randomly sampled** for training.
>
> This training budget is **comparable to prior academic works** on GRPO training [1,2,3], which typically adopt a limited number of steps (around or fewer than 1,000). In our experiments, we find that such a training budget already achieves strong performance while remaining affordable for conducting ablation studies. Meanwhile, we believe the larger dataset we provide can serve as a valuable resource for future research on large-scale RL training. Due to computational and time constraints, we leave the exploration of RL training with large-scale data to future work.
>
> We will clarify these points in the final version.
>
> [1] VideoRFT: Incentivizing Video Reasoning Capability in MLLMs via Reinforced Fine-Tuning
>
> [2] VLM-R1: A Stable and Generalizable R1-style Large Vision-Language Model
>
> [3] Search-R1: Training LLMs to Reason and Leverage Search Engines with Reinforcement Learning

---

> ### Author Response · Authors · 2025-08-05
>
> > Q3. Are SophiaVL-R1-130k and SophiaVL-R1-Thinking-156k derived from the same source data? Could the authors provide a textual explanation or a table to clarify the data sources and composition of the SophiaVL-R1-Thinking-156k dataset?
>
> A3: Thanks for your comment. Yes, **SophiaVL-R1-Thinking-156k** is constructed based on **SophiaVL-R1-130k**, by collecting multiple reasoning responses of each sample during the GRPO training of Qwen2.5-VL-7B-Instruct model on the **SophiaVL-R1-130k** dataset.
>
> During the GRPO training, we collect all generated responses and **filter out low-quality samples**, such as those that only provide a final answer without intermediate reasoning. Notably, around 40% of the responses are identified as low-quality and are mostly removed.
>
> Next, we use **Qwen2.5-VL-72B-Instruct** to annotate the remaining samples, assigning reward score to each response. However, as shown in Figure 2 (right), the distribution across reward intervals was highly imbalanced. To fix this, we select a specific number of samples from each reward interval, keeping the count between 5000 and 15000. This ensures a more balanced distribution across different reward levels. The resulting dataset is our final **SophiaVL-R1-Thinking-156k**, whose score distribution is shown as **'Selected'** in Figure 2 (right) of the manuscript.
>
> We apologize for omitting this explanation in the main paper. For clarity, we provide a summary table below that outlines the composition of SophiaVL-R1-Thinking-156k.
>
> | Division                     | Data source       | # Reasoning Responses |
> | - | - | - |
> | Knowledge (Total: 26231) | TQA               | 3278                  |
> |                              | PMC-VQA           | 5587                  |
> |                              | EXAMS-V           | 2134                  |
> |                              | viquae            | 1639                  |
> |                              | AI2D              | 3097                  |
> |                              | ScienceQA         | 4016                  |
> |                              | ArxivQA           | 4683                  |
> |                              | VQA-RAD           | 452                   |
> |                              | GVLQA             | 1078                  |
> |                              | AI2D-gpt4v        | 258                   |
> | OCR (Total: 36402)           | Rendered_Text     | 5987                  |
> |                              | IAM               | 5197                  |
> |                              | HME100k           | 13454                 |
> |                              | TextCaps          | 3714                  |
> |                              | TextVQA           | 3644                  |
> |                              | TextOCR           | 3383                  |
> |                              | ChromeWriting     | 1023                  |
> | General (Total: 14827)       | A-OKVQA           | 4963                  |
> |                              | Visual7W          | 1258                  |
> |                              | ShareGPT4o        | 4962                  |
> |                              | IconQA            | 2054                  |
> |                              | ShareGPT4V        | 1590                  |
> | Chart (Total: 33527)         | TabMWP            | 4855                  |
> |                              | FigureQA          | 2392                  |
> |                              | RoBUT_SQA         | 4874                  |
> |                              | MapQA             | 1879                  |
> |                              | DVQA              | 3346                  |
> |                              | PlotQA            | 1829                  |
> |                              | VisualWebInstruct | 9274                  |
> |                              | ChartQA           | 1756                  |
> |                              | Chart2Text        | 3322                  |
> | Math (Total: 45716)          | GeomVerse         | 3450                  |
> |                              | Multimath-300k    | 23105                 |
> |                              | UniGeo            | 4923                  |
> |                              | Geometry3K        | 2917                  |
> |                              | GeoQA+            | 1753                  |
> |                              | CLEVR-Math        | 986                   |
> |                              | MAVIS-Geometry    | 6304                  |
> |                              | Super-CLEVR       | 1339                  |
> |                              | GEOS              | 203                   |
> |                              | LIMO              | 608                   |
> |                              | MetaMathQA        | 128                   |
>
> We will turn this table into a figure in the final version of this paper.

---

> ### Author Response · Authors · 2025-08-05
>
> > Q4. It is also important to provide an evaluation of the process quality, especially since the authors used a reward model to supervise the intermediate reasoning steps. Moreover, I believe that evaluating on MME-CoT would not be time-consuming.
>
> A4: Thanks for your comment. We have completed the evaluation on MME-CoT after the rebuttal submission, and now report the results below:
>
> |                        | CoT Quality |  |  | CoT Robustness |  |  | CoT Efficiency |  |  |
> |-|-|-|-|-|-|-|-|-|-|
> |                        | **F1 Score** ($\uparrow$) | Precision ($\uparrow$) | Recall ($\uparrow$) | **Avg. Score** ($\uparrow$) | Stability ($\uparrow$) | Efficacy ($\uparrow$) | **Avg. Score** ($\uparrow$) | Relevance rate ($\uparrow$) | Reflection quality ($\uparrow$) |
> | Qwen2-VL-7B            | 42.1        | 61.6 | 32.0 | -4.0 | -3.1 | -4.8 | 94.9 | 89.8 | 100 |
> | LLaVA-OV-72B           | 36.3        | 57.3 | 26.6 | -0.2 | 0.3  | -0.6 | 94.0 | 88.1 | 100 |
> | Qwen2.5-VL-7B-Instruct | 41.1        | 72.4 | 28.7 | -0.3 | -1.0 | 0.5  | 58.4 | 86.8 | 30.0 |
> | SophiaVL-R1            | 43.3        | 77.6 | 30.1 | 0.2  | 0.0  | 0.4  | 82.5 | 88.6 | 76.4 |
> 1. **CoT Quality and Robustness Gains.** SophiaVL-R1 achieves superior performance in `CoT quality`, especially in terms of `precision`, indicating its intermediate reasoning steps are more aligned with ground-truth rationales. It also shows stronger `CoT robustness` compared to baselines.
>
> 2. **Efficiency Metric Clarification.** Although SophiaVL-R1 scores *lower* than Qwen2-VL-7B and LLaVA-OV-72B in `CoT efficiency`, the difference is mainly caused by how `reflection quality` is computed in MME-CoT. **Specifically, it is difficult to directly compare models with and without reflection capability, as they are evaluated using different criteria.** Detailed analysis is shown below.
>
>     Let $ \mathcal{R} $ denote the set of all reflection steps identified by gpt-4o from the model’s responses to MME-CoT questions. Let $ \mathcal{R}_{\text{valid}} $ be the subset of $\mathcal{R}$ that are considered valid—i.e., those that either correctly identify prior mistakes or provide new insights to verify previous conclusions. Then, the `reflection quality` score is computed as:
> $$
> \text{Reflection Quality} =
> \begin{cases}
> \displaystyle \frac{|\mathcal{R}_{\text{valid}}|}{|\mathcal{R}|} \times 100, & \text{model supports reflection} \\\\
> 100, & \text{model does not support reflection}
> \end{cases}
> $$
>
>   - **For models incapable of reflection** (i.e., gpt-4o detects no reflection steps in their responses, $|\mathcal{R}|$ = 0), MME-CoT assigns a default reflection quality score of 100, as stated in Section 4.2 of the original paper [1]. Examples include Qwen2-VL-7B and LLaVA-OV-72B.
>
>    - **For models capable of reflection** (e.g., SophiaVL-R1), the reflection quality score is computed as the proportion of valid reflection steps.
>
> This leads to an **incomparable metric** range between models with and without reflection capability.
>
> In some cases, however, models not explicitly designed for reflection may still exhibit reflection-like behavior in their responses. We observe that Qwen2.5-VL-7B-Instruct receives a `reflection quality` score of 30.0, despite not being explicitly designed for reflection. By manually checking its outputs, we find Qwen2.5-VL-7B-Instruct sometimes includes phrases indicating reflections. (e.g., *"However, upon re-evaluation..."* or *"This result does not match any of the provided options. Let's re-evaluate the problem by..."*), which are detected as reflections $\mathcal{R}$ by the evaluator model (gpt-4o).
>
> In contrast, after being trained with our proposed method, Qwen2.5-VL-7B-Instruct is significantly improved, resulting in SophiaVL-R1. **Compared to the original base model, SophiaVL-R1 shows much higher reflection quality and average CoT efficiency**, indicating not only better reflection capabilities but also more effective use of reflection to enhance reasoning performance.
>
> Overall, these results support the effectiveness of enhancing reasoning capability of SophiaVL-R1, which leads to improved thinking quality in chain-of-thought reasoning.
>
> [1] MME-CoT: Benchmarking Chain-of-Thought in Large Multimodal Models for Reasoning Quality, Robustness, and Efficiency
>
> > Q5. Since Figure 5 only includes rule-based outcome reward, why do the initial performance levels of the methods shown in the figure not start from the same baseline?
>
> A5: Thanks for your comment. The differences in initial performance arise from the stochastic nature of the GRPO algorithm, which samples multiple responses per question as rollouts. Specifically, we use top-p sampling ($p=0.9$) with a temperature of 1.0 to encourage response diversity during training. As a result, the model produces diverse responses with varying average outcome rewards for each question in the batch, leading to differences in initial training rewards.

---

> > ### Comment · Reviewer_XUT8 · 2025-08-08
> >
> > Thank you for your reply. My question has been resolved, and I will raise the rating to 5.

---

### Official Review · Reviewer_tjSX · 2025-07-02

**Clarity:** 4
**Significance:** 3
**Originality:** 3
**Rating:** 5
**Confidence:** 3

**Summary:**

This paper introduces SophiaVL-R1, a multi-modal large language model designed to improve reasoning abilities by incorporating "thinking rewards" during reinforcement learning training. The core idea is to supervise the model's thinking process, not just its final outcome. To address potential unreliability in these thinking rewards, the authors propose Trust-GRPO, an algorithm that assigns a trustworthiness weight to the thinking reward. They also use an annealing strategy to gradually reduce its influence over time, favoring the more accurate rule-based outcome reward in later stages of training. Experiments demonstrate that SophiaVL-R1 achieves strong performance on various reasoning benchmarks, even outperforming larger models.

**Questions:**

1.  DeepSeek-R1 focuses on outcome rewards. Why is process supervision now effective, and is its effectiveness tied to the base model's size or intrinsic capabilities? Clarifying this connection would strengthen the argument for process supervision.
2.  In Figure 5, the training curves show smoother behavior at the beginning and end, but are noticeably rough in the middle. Can the authors explain the reason for this "jagged" pattern in the middle section of the curves?
3.  The main experiments use Qwen2.5-VL-7B-Instruct as the base model. Do the authors believe the findings regarding Trust-GRPO are universally applicable to other MLLMs, or are they specific to the Qwen family of models?

**Ethical Concerns:**

["NO or VERY MINOR ethics concerns only"]

**Final Justification:**

Thank the authors for their detailed explanations. It is a strong work both for academic and industrial. I’d like to maintain my score and suggest acceptance.

**Limitations:**

Yes, the authors have adequately addressed the limitations.

**Quality:**

3

**Strengths And Weaknesses:**

**Strengths:**

1. The paper tackles a crucial problem in MLLM reasoning: the lack of supervision over the thinking process.
2. The literature review appears comprehensive, covering relevant work on process reward models and MLLM reasoning.
3. The motivation for adding thinking rewards is well-justified. The paper clearly explains that relying solely on outcome rewards can lead to suboptimal reasoning strategies. Figure 1 effectively illustrates this point.
4. The method seems sound. Training a dedicated thinking reward model and then integrating it into the GRPO framework with mechanisms to handle potential reward unreliability (Trust-GRPO) is a commonly used approach.
5. The empirical evaluations are of high quality. The paper tests SophiaVL-R1 on a diverse set of standard benchmarks, and compares it against several strong baselines.
6. The results are significant. SophiaVL-R1-7B consistently outperforms existing MLLMs on various benchmarks. The computational resources needed are not excessively high for a model of this capability.
7. The paper is clearly written and easy to understand.
8. The authors have included a discussion of limitations, which is appreciated.

**Weaknesses:**

1. DeepSeek-R1 focuses on outcome rewards. It would be beneficial to discuss why thinking supervision is effective now. Is it related to model size or inherent model capabilities?
2. The training curves in Figure 5 show smooth beginnings and ends, but the middle sections are "jagged" or uneven. An explanation for this behavior would enhance the clarity of the results.
3. While the experiments are strong on Qwen2.5-VL-7B-Instruct, it's unclear if the conclusions (e.g., the effectiveness of Trust-GRPO) generalize to other base MLLMs.

---

> ### Author Rebuttal · Authors · 2025-07-30
>
> > Q1: DeepSeek-R1 focuses on outcome rewards. It would be beneficial to discuss why thinking supervision is effective now. Is it related to model size or inherent model capabilities?
> >
> > DeepSeek-R1 focuses on outcome rewards. Why is process supervision now effective, and is its effectiveness tied to the base model's size or intrinsic capabilities? Clarifying this connection would strengthen the argument for process supervision.
>
> A1: Thanks for your comments.
>
> 1. **Why our thinking process supervision is effective**
>
> Thanks for your advice. We find that the "wrong thinking, correct answer" phenomenon exists in R1-style training, which may cause the model to learn flawed reasoning strategies and ultimately reduce its generalization ability. A recent work [1] (July 2025) in the industry also reveals this challenge: "although RL enhances task completion rates, it does not consistently improve reasoning quality", "flawed or hallucinated reasoning chains may inadvertently be reinforced if they produce correct answers."
>
> However, previous approaches like PRM may struggle to deal with this due to two may reasons: (1) Rigid step-wise constraints may hinder flexibility and reduce generalization, especially on general tasks (2) evaluating step correctness is difficult, which may lead models to exploit rewards by repeating or adding meaningless steps. In the experiment, one of our PRM-based baseline models, InternVL2.5-8B-VisualPRM, demonstrates inferior performance in Tables 1 and 2 of our manuscript, indicating the limitations of PRM-based approaches.
>
> In contrast, our thinking reward model evaluates the entire reasoning trace holistically, considering aspects like coherence and correctness, rather than scoring each step individually. This leads to more reliable supervision. Additionally, our Trust-GRPO method incorporates trustworthiness weighting and time-based annealing to reduce the impact of noisy or hacked rewards. Together, these designs make thinking process supervision both more robust and practical for today’s models.
>
> [1] GLM-4.1V-Thinking: Towards Versatile Multimodal Reasoning with Scalable Reinforcement Learning
>
> 2. **Its effectiveness tied to the base model's size or intrinsic capabilities**
>
> The effectiveness of thinking-level reward primarily stems from its ability to mitigate the "wrong reasoning, correct answer" phenomenon. In such cases, the model receives high rewards despite flawed reasoning, which may mislead learning and hinder generalization.
>
> This issue is inherent to the design of outcome-based RL algorithm such as GRPO, which focus solely on outcome rewards without assessing the reasoning process. Therefore, the benefit of thinking reward is not tied to model size—both small and large models could be affected by this problem.
>
> We believe that applying our method to larger models may also bring benefits. Due to time constraints during the rebuttal phase, we have not yet conducted such experiments, but we will explore the effectiveness of our method on larger models in the future.
>
> > Q2: The training curves in Figure 5 show smooth beginnings and ends, but the middle sections are "jagged" or uneven. An explanation for this behavior would enhance the clarity of the results.
> >
> > In Figure 5, the training curves show smoother behavior at the beginning and end, but are noticeably rough in the middle. Can the authors explain the reason for this "jagged" pattern in the middle section of the curves?
>
> A2: Thanks for your comments. In the early stage of training, the model may primarily rely on its pre-trained reasoning abilities, resulting in relatively smooth and stable performance curves.
>
> During the middle stage, we guess that the model begins to explore alternative reasoning strategies in order to maximize rewards. This exploration phase likely introduces instability and fluctuations in performance, which explains the jagged pattern observed in the curves.
>
> In the later stage, as the model converges on more effective reasoning strategies, its behavior stabilizes again, leading to smoother training curves toward the end of the process.
>
> > Q3: While the experiments are strong on Qwen2.5-VL-7B-Instruct, it's unclear if the conclusions (e.g., the effectiveness of Trust-GRPO) generalize to other base MLLMs. Q3. The main experiments use Qwen2.5-VL-7B-Instruct as the base model. Do the authors believe the findings regarding Trust-GRPO are universally applicable to other MLLMs, or are they specific to the Qwen family of models?
>
> A3: Thanks for your comments. We choose Qwen2.5-VL-7B-Instruct as the base model mainly because it has recently become a popular foundation for open-source multimodal instruction tuning, as demonstrated by recent works such as Visual-RFT [1], R1-OneVision [2], and Reason-RFT [3]. Following this line of research allows for direct comparison and fair benchmarking.
>
> In principle, our method—Trust-GRPO—is a modification of the GRPO training framework itself and does not rely on any architecture or optimization specific to the Qwen family. Therefore, we believe it can be applied to other multimodal large language models as well. However, due to limited time and computational resources, we leave this exploration to future work.
>
> [1] Visual-RFT: Visual Reinforcement Fine-Tuning
>
> [2] R1-Onevision: Advancing Generalized Multimodal Reasoning through Cross-Modal Formalization
>
> [3] Reason-RFT: Reinforcement Fine-Tuning for Visual Reasoning

---

### Official Review · Reviewer_VCHF · 2025-07-13

**Clarity:** 3
**Significance:** 3
**Originality:** 2
**Rating:** 5
**Confidence:** 4

**Summary:**

This paper presents SophiaVL-R1, an MLLM trained with a new reinforcement learning (RL) approach called Trust-GRPO, which combines rule-based outcome rewards and holistic thinking rewards. The core contribution is the introduction of a thinking reward model that evaluates the intermediate reasoning process, not just the final answer, and a trustworthiness mechanism to mitigate unreliable reward signals. Key points include: 1) A holistic-level thinking reward model trained on annotated multimodal reasoning samples. 2) The Trust-GRPO algorithm, which down-weights misleading rewards using a group-based trustworthiness weight. 3) An annealing strategy to decay the influence of thinking rewards over time.

Experiments on MathVista, MMMU, MathVerse, and other benchmarks demonstrate competitive performance. Remarkably, SophiaVL-R1-7B outperforms models with much larger parameter counts.

**Questions:**

- Can the authors justify the choice of exponential decay for the annealing schedule? Would a learned schedule or reward-gated strategy perform better?
- Could the trustworthiness weight be refined beyond mean reward comparison? For example, using variance, entropy, or confidence intervals?
- Are there signs of overfitting to the benchmarks used? The impressive results on known datasets might not generalize to unseen or adversarial tasks.

**Ethical Concerns:**

["NO or VERY MINOR ethics concerns only"]

**Final Justification:**

After considering the response, I am increasing my score from 4 to 5.
- The authors addressed my main concerns by providing extra evaluations on unseen tasks (MMT-Bench), which alleviates worries about overfitting to known benchmarks.
- The authors also demonstrated that alternative annealing schedules (linear vs. exponential) yield similar gains, reinforcing that the core idea is robust.
Some design choices remain heuristic and could benefit from deeper theoretical justification, but overall the work is technically solid, impactful for the MLLM reasoning community, and supported by convincing empirical results.

**Limitations:**

yes

**Quality:**

3

**Strengths And Weaknesses:**

Strengths:
- The paper targets a real problem: reward hacking and unreliable intermediate reasoning in MLLMs trained via RL.
- The combination of a holistic thinking reward, trustworthiness weight, and annealing within RL is novel and well-integrated.
- SophiaVL-R1-7B achieves superior performance on a variety of MLLM benchmarks, outperforming models like LLaVA-OneVision-72B.
- Implementation details are well-documented, and the authors promise to release all data and code.
- Components like trust weighting and annealing are evaluated and shown to contribute meaningfully.

Weaknesses:
- While useful, the paper extends an existing GRPO pipeline with seemingly intuitive tweaks (reward model, trust weight, annealing). The novelty might be considered limited.
- Trustworthiness metric is heuristic. The formulation of the weight (trustworthiness) based on group averages is simple and not deeply justified beyond empirical behavior.
- The decision to decay thinking rewards over time seems heuristic. There is no theoretical or empirical exploration of alternative scheduling strategies.

---

> ### Author Rebuttal · Authors · 2025-07-30
>
> > Q1: While useful, the paper extends an existing GRPO pipeline with seemingly intuitive tweaks (reward model, trust weight, annealing). The novelty might be considered limited.
>
> A1: Thanks for your comments. This work represents an early attempt to address the common “wrong reasoning, correct answer” phenomenon in the R1-style training of MLLMs.
>
> To address this, we curated two diverse and carefully constructed datasets:  SophiaVL-R1-130k for reasoning MLLM training, and SophiaVL-R1-Thinking-156k specifically designed to train the thinking reward model tailored for GRPO. We believe these datasets provide valuable resources to the community for improving reasoning supervision beyond final outcome rewards.
>
> Alongside this, we propose a lightweight trustworthiness weighting mechanism and a time-based annealing strategy that improve training efficiency and stability without extra computational cost, offering practical approaches to advance multimodal reasoning models.
>
> We hope our work provides valuable insights and a practical framework that can inspire further research in the R1-style training of MLLMs.
>
> > Q2: Trustworthiness metric is heuristic. The formulation of the weight (trustworthiness) based on group averages is simple and not deeply justified beyond empirical behavior.
> >
> > Could the trustworthiness weight be refined beyond mean reward comparison? For example, using variance, entropy, or confidence intervals?
>
> A2: Thanks for your comments. Our goal in designing the trustworthiness weight is to provide a simple and efficient estimation tailored to GRPO, without introducing additional computational cost—this is important given the high cost of training and inference in MLLMs.
>
> We also explore an alternative method using variance-based uncertainty: for each model response, we independently evaluate it three times using the thinking reward model to obtain three thinking reward scores (denoted as $r_1, r_2, r_3$), and compute their variance. A higher variance indicates greater uncertainty, and thus a lower trustworthiness. The trustworthiness weight $\gamma$ is defined as follows:
> $$
> \gamma = \exp\left(-\frac{1}{3} \sum_{i=1}^{3} \left(r_i - \frac{1}{3} \sum_{j=1}^{3} r_j\right)^2\right)
> $$
> The results are listed in the following table.
>
> |                        | MathVista (Math) | MMBench (General) |
> | ---------------------- | ---------------- | ----------------- |
> | Qwen2.5-VL-7B-Instruct | 67.5             | 83.3              |
> | SophiaVL-R1 (variance) | 69.1             | 85.1              |
> | SophiaVL-R1            | 71.3             | 85.4              |
>
> However, as shown in the table, the variance-based approach  underperforms our method, while requiring more computation. This demonstrates that our method is both effective and efficient in practice.
>
> > Q3: The decision to decay thinking rewards over time seems heuristic. There is no theoretical or empirical exploration of alternative scheduling strategies.
> >
> > Can the authors justify the choice of exponential decay for the annealing schedule? Would a learned schedule or reward-gated strategy perform better?
>
> A3: Thank you for your comments. The idea of time-based decay is to gradually reduce the influence of the thinking reward as training progresses, encouraging the model to rely more on outcome rewards in later stages, where they tend to be more accurate and stable. This strategy helps balance early guidance from intermediate reasoning signals with the eventual goal of producing correct responses.
>
> In principle, the specific form of decay is not critical. We also experiment with a linear decay schedule, which achieves comparable improvements (see table below). This supports our core idea that decaying the thinking reward over time is beneficial, regardless of the exact function used.
>
> |                            | MathVista (Math) | MMBench (General) |
> | -------------------------- | --------- | ------- |
> | Qwen2.5VL-7B-Instruct      | 67.5      | 83.3    |
> | SophiaVL-R1 (linear decay) | 70.2      | 84.1    |
> | SophiaVL-R1                | 71.3      | 85.4    |
>
> We agree that more adaptive strategies, such as learned or reward-gated schedules, may further improve performance, and we plan to explore them in future work.
>
> > Q4: Are there signs of overfitting to the benchmarks used? The impressive results on known datasets might not generalize to unseen or adversarial tasks.
>
> A4: Thanks for your comments. To further verify the generalization ability of our model, we additionally evaluate SophiaVL-R1 on **MMT-Bench** **(val)**, a comprehensive multimodal benchmark covering 162 tasks, including many categories **unseen during training**, such as Localization and GUI Navigation. Results on unseen tasks are as follows:
>
> |                        | All tasks | Localization task | Temporal Understanding task | GUI Navigation task | Cross Image Matching task |
> | ---------------------- | --------- | ----------------- | --------------------------- | ------------------ | ------------------------ |
> | Qwen2.5-VL-7B-Instruct | 61.9      | 39.1              | 43.5                        | 30.0               | 35.0                     |
> | SophiaVL-R1            | 62.7      | 46.9              | 51.5                        | 35.3               | 43.3                     |
>
> From the above table, we can find that SophiaVL-R1 achieves better performance than Qwen2.5-VL-7B-Instruct, indicating that our model generalizes well beyond the training tasks.

---

> > ### Comment · Reviewer_VCHF · 2025-08-09
> > **Thanks for your response**
> >
> > Thank you for the detailed clarifications and additional experiments. The extra evaluations on unseen tasks and the comparison with variance-based trustworthiness weighting help address my concerns on generalization and the heuristic nature of your design choices. I also appreciate the release of carefully constructed datasets, which strengthens the contribution beyond incremental algorithmic changes. Based on these clarifications and results, I am raising my score to 5.

---

### Decision · Program_Chairs · 2025-09-17

**Decision:**

Reject

**Comment:**

This paper presents SophiaVL-R1, a multimodal large language model that enhances reasoning capabilities through Trust-GRPO, which combines rule-based outcome rewards with holistic thinking rewards. The method aims to address the "wrong reasoning, correct answer" phenomenon in R1-style training.

There are several fundamental concerns merit closer examination:
1. Limited practical improvements over baseline. The paper claims that SophiaVL-R1-7B outperforms LLaVA-OneVision-72B, but the most important comparison in this paper should be between Trust-GRPO and standard GRPO using the same base model. When properly compared using Qwen2.5-VL-7B-Instruct, Trust-GRPO shows only modest improvements of 1-2 absolute points over GRPO on 4 out of 7 benchmarks. These marginal gains raise questions about the practical significance of the proposed method.
2. The thinking rewards appear to provide limited benefits, requiring both a trustworthiness metric and exponential decay strategy to be effective. Furthermore, the linear decay experiment presented during rebuttal—which would upweight thinking rewards more than exponential decay throughout training—resulted in degraded performance.
3. Lack of quantitative reliability analysis. The paper provides no rigorous quantitative analysis of thinking reward reliability. While the authors presented Spearman correlation (0.55) with GPT-4o annotations during rebuttal, this metric is difficult to interpret without knowing GPT-4o's own reliability.
4. Presentation issues. Table 3's first row appears to incorrectly present base model results without GRPO training (copied from Tables 1-2) rather than actual GRPO baseline results.

Given the concerns above, the recommendation is to reject this paper.